# ROBUST STEREO MATCHING BY RISK MINIMIZATION

## ABSTRACT

This paper presents a novel formulation for capturing the continuous disparity in stereo matching networks. In contrast to previous approaches that regress the final output as the expectation of discretized disparity values, we derive a continuous modeling formulation by treating the predicted disparity as an optimal solution to the risk minimization problem. We demonstrate that the commonly used disparity expectation represents an $L^2$ special case within the proposed risk formulation, and transitioning to an $L^1$ formulation notably enhances stereo matching robustness, particularly for disparities with multi-modal probability distributions. Moreover, to enable the end-to-end network training with the non-differentiable $L^1$ risk optimization, we explored the well-known implicit function theorem and proposed a differentiable scheme for both network forward prediction and backward propagation. A comprehensive analysis of our proposed formulation demonstrates its theoretical soundness and superior performance over current state-of-the-art methods across various benchmarks, including KITTI 2012, KITTI 2015, ETH3D, SceneFlow, and Middlebury 2014. We believe our work not only advances the field of stereo matching but also holds promise for broader applications, spanning computer vision, robotics, and control engineering.

## 1 INTRODUCTION

Stereo Matching is one of the most important and fundamental problems in computer vision (Hoff & Ahuja, 1989; Kang et al., 1995; Scharstein & Szeliski, 2002; Szeliski, 2022). Given a rectified stereo image pair captured at the same timestamp, the goal of stereo matching is to estimate the per-pixel displacement from left to right images, popularly known as a disparity map. Under the rectified image pair setup, the stereo matching problem boils down to a well-structured 1D search problem in the image space (Szeliski, 2022). Due to its effectiveness and affordability, stereo camera rigs have been widely adopted in commercial and industrial applications, including autonomous driving cars (Fan et al., 2020; Bimbraw, 2015), smartphones (Meuleman et al., 2022; Luo et al., 2020; Pang et al., 2018), and other robotic automation systems (Kim et al., 2021; Hsieh & Lin, 2020).

Classical well-known stereo matching methods—often categorized as local methods, use a predefined support window to find suitable matches between stereo image pair (Scharstein & Szeliski, 2002; Hirschmuller, 2007). Yet, approaches that optimize for all disparity values using a global cost function were observed to provide better results (Kolmogorov & Zabih, 2001; Klaus et al., 2006; Bleyer et al., 2011; Yamaguchi et al., 2014). In recent years, with the surge in high-quality, large-scale synthetic ground-truth data, availability of high-end GPUs' and advancements in deep-learning architecture, the neural network-based stereo matching models trained under supervised setting has outperformed classical methods accuracy by a significant margin (Kendall et al., 2017a; Chang & Chen, 2018; Zhang et al., 2019; Lipson et al., 2021). Nevertheless, one fundamental challenge still remains, i.e. how to model *continuous* scene disparity values given only a limited number of candidate pixels to match? After all, the scene is continuous in nature.

Many recent works have attempted to overcome the above challenge of predicting continuous scene disparities, which can be broadly divided into two categories. *(i)* **Regression-based approaches** predict a real-valued offset by neural networks for each hypothesis of discrete disparity. The offset is then added to the discrete disparity hypothesis as the final continuous prediction. Typical examples include RAFT-Stereo (Lipson et al., 2021), CDN (Garg et al., 2020), and more recent IGEV (Xu et al., 2023) and DLNR (Zhao et al., 2023). *(ii)* **Classification-based approaches** first esti-

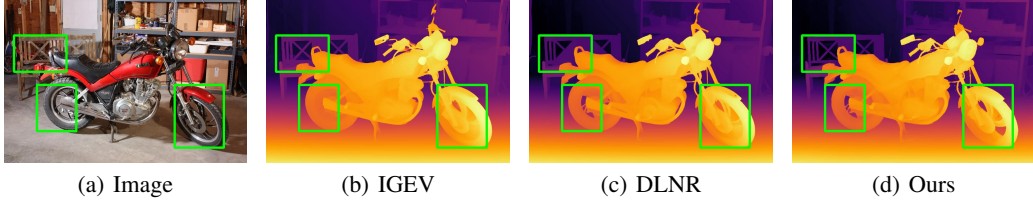

|  |  |  |  |
|:---:|:---:|:---:|:---:|
| (a) Image | (b) IGEV | (c) DLNR | (d) Ours |

Figure 1: **Qualitative Comparison.** We compare our method with recent state-of-the-art methods such as IGEV (Xu et al., 2023), DLNR (Zhao et al., 2023) on Middlebury (Scharstein & Szeliski, 2002). All methods are trained only on SceneFlow (Mayer et al., 2016), and evaluated at quarter resolution. It can be observed that our method generalizes and predicts high-frequency details better than other recent methods.

mate the categorical distribution[1] for the discrete disparity hypotheses and then take the expectation value of the distribution as the final disparity, which can be any arbitrary real value even though the categorical distribution is discrete (Kendall et al., 2017a; Chang & Chen, 2018; Zhang et al., 2019).

In this paper, we aim to address the importance of continuous disparity modeling in stereo matching, given the categorical distribution of disparity hypotheses. We introduce a radically different perspective on the disparity prediction problem by framing it as a search problem of finding the minimum risk (Lehmann & Casella, 1998; Vapnik, 1991; Berger, 2013) of disparity values. Specifically, the risk is defined by averaging the prediction error with respect to all possible values of the ground-truth disparity. At the time of making the prediction, the ground truth is unavailable, which is therefore approximated by the disparity hypotheses with a categorical distribution. We search for a disparity value as our prediction that achieves minimal overall risk involved with it. Moreover, we demonstrate that the commonly used disparity expectation (Kendall et al., 2017a) is a special case of $L^2$ error function within the proposed risk formulation, which is sensitive to multi-modal distribution and may result in the over-smooth solution (Chen et al., 2019; Tosi et al., 2021). In contrast, we advocate the use of the $L^1$ error function during risk minimization.

Despite the theoretical soundness of the $L^1$ risk minimization, there is no closed-form solution to $L^1$ formulation. To that end, in this paper, we search for the solution by computing derivatives of our proposed risk function and performing its continuous optimization. By interpolating the disparity categorical distribution, we define our continuous probability density function. Then, we propose a binary search algorithm to find the optimal disparity that minimizes the proposed risk efficiently. To enable the end-to-end network training, we compute the backward gradient of the final disparity with respect to the categorical distribution by the implicit function theorem (Krantz & Parks, 2002).

We have extensively evaluated the proposed method on a variety of stereo matching datasets. Our approach demonstrates superior performance compared to many state-of-the-art methods on benchmarks such as SceneFlow (Mayer et al., 2016), KITTI 2012 (Geiger et al., 2012), and KITTI 2015 (Menze & Geiger, 2015). Moreover, our approach achieves significantly better cross-domain generalization, as observed on Middlebury (Scharstein & Szeliski, 2002), ETH 3D (Schöps et al., 2017), KITTI 2012 & 2015. An example of qualitative comparison is given in Fig. 1. Ablation studies confirm the effectiveness of risk minimization, not only within the proposed network but also in the context of general stereo matching networks, such as ACVNet (Xu et al., 2022) and PCWNet (Shen et al., 2022).

## 2 RELATED WORK

### 2.1 DEEP NEURAL NETWORK FOR STEREO MATCHING

In recent years, the deep-learning based approaches have improved the accuracy of stereo matching by a significant margin. Designing powerful and efficient network architectures for stereo matching is a popular research topic. Zbontar & LeCun (2015) apply deep convolutional networks (LeCun

---

[1]A categorical distribution is a discrete probability distribution that describes the possible results of a random variable that can take on the K possible categories, with the probability of each category separately specified.

et al., 1995) to learn discriminative features for image patches. DispNetCorr (Mayer et al., 2016) designs explicit correlation in networks to construct cost volume. GCNet (Kendall et al., 2017a) constructs volume by concatenation and refines by 3D convolution. PSM-Net (Chang & Chen, 2018) exploits spatial pyramid pooling (Zhao et al., 2017) and stacked hourglass (Newell et al., 2016) to learn context information. STTR (Li et al., 2021) applies transformers (Vaswani et al., 2017; Dosovitskiy et al., 2021) to relax the limitation of a fixed disparity range. Moreover, the uniqueness constraint is considered by optimal transport (Cuturi, 2013). ACVNet (Xu et al., 2022) weights the matching costs by attention.

Another line of research is to improve efficiency. In GANet (Zhang et al., 2019) the computationally costly 3D convolutions are replaced by the differentiable semi-global aggregation (Hirschmuller, 2007). GWCNet (Guo et al., 2019) constructs the cost volume by group-wise correlation. AANet (Xu & Zhang, 2020) proposes the adaptive cost aggregation to replace the 3D convolution for efficiency. AnyNet (Wang et al., 2019), DeepPruner (Duggal et al., 2019), HITNet (Tankovich et al., 2021), CasMVSNet (Gu et al., 2020), PCWNet (Shen et al., 2022) and Bi3D (Badki et al., 2020) prune the range of disparity in the iterative manner. RAFT-Stereo (Lipson et al., 2021), CREStereo(Li et al., 2022), IGEV (Xu et al., 2023) and DLNR (Zhao et al., 2023) use recurrent neural networks (Cho et al., 2014) to predict and refine the disparity iteratively.

In this paper, our network structure is inspired by CasMVSNet (Gu et al., 2020), and consists of two stages to predict and refine the disparity map. The hierarchical design reduces the time and memory cost, while keeping the matching accuracy.

## 2.2 CONTINUOUS DISPARITY BY CLASSIFICATION

In deep networks that have cost volumes, the most popular way to predict the disparity from the volume is the weighted average operation, i.e. expectation. Chen et al. (2019) find the average operation suffers from the over-smoothing problem, especially at the boundaries of objects. Therefore they propose the single-modal weighted average. Garg et al. (2020) propose to predict a continuous offset to shift the distribution modes of disparity. Furthermore, they generate multi-modal ground truth disparity distributions and supervise the network to learn the distribution by Wasserstein distance (Villani, 2008). SMD-Net (Tosi et al., 2021) exploit bimodal mixture densities as output representation for disparities. UniMVSNet (Peng et al., 2022) attempts to unify the advantages of classification and regression by designing a novel representation, and further proposes a unified focal loss. Yang et al. (2022) use top-K hypotheses for the disparity to alleviate the multi-modal problem. In this paper, we propose to minimize the risk under $L^1$ norm to capture continuous disparity and solve the multi-modal problem. Moreover, our approach can be trained in an end-to-end manner.

## 2.3 CROSS-DOMAIN GENERALIZATION

Existing real-world stereo datasets are small and insufficient to train neural networks from scratch, therefore exploiting synthetic images to pre-train networks and reducing the domain gap play an important role. Tonioni et al. (2017; 2019a;b) fine tune the stereo matching networks on the target domain using unsupervised loss. Liu et al. (2020) jointly optimize networks for domain translation and stereo matching during training. Zhang et al. (2020); Song et al. (2021) normalize features to reduce domain shifts. Cai et al. (2020); Liu et al. (2022a) design robust features for stereo matching. Liu et al. (2022b) find the cost volume built by cosine similarity generalizes better to different image features. Zhang et al. (2022) apply the stereo contrastive loss and selective whitening loss to improve feature consistency. Chang et al. (2023) proposed the hierarchical visual transformation to learn shortcut-invariant robust representation from synthetic images. In this paper, we present a novel perspective to improve robustness by $L^1$ risk minimization. We also show that our approach can be combined with above methods to further improve the robustness.

## 3 METHOD

### 3.1 PROBABILITY DENSITY OF CONTINUOUS DISPARITY

For each pixel in the left image, suppose the possible disparities are in the range of $[d_{\min}, d_{\max}]$. Typical stereo matching algorithms will compute a cost that merely can be described as a probabil-

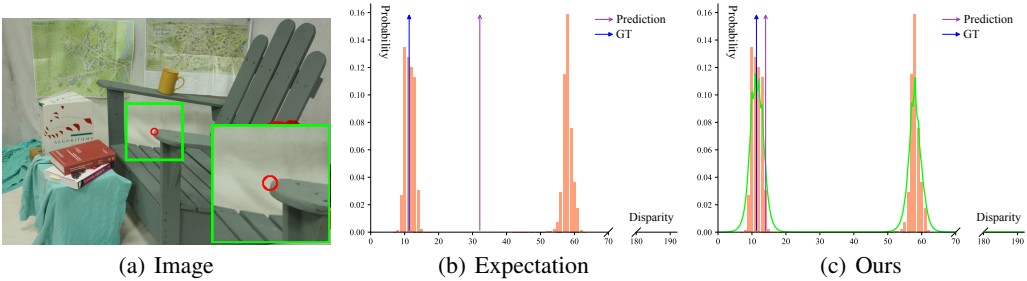

|      (a) Image      |      (b) Expectation      |      (c) Ours      |

Figure 2: **Illustration of the difference between the expectation and our method.** In (a) the pixel in the red circle is located at the boundary of the chair, thereby the distribution of the disparity has multiple modes. We plot the discrete distribution of disparity hypotheses by orange bars in (b) and (c). In (b) the prediction obtained by averaging is blurred and far from any of the modes. In (c) we find the optimal solution under $L^1$ norm, which is more robust and closer to the ground truth. The green curve is the interpolated probability density.

ity mass function (PMF) with a finite set of disparities $\mathbf{d} = [d_1, ..., d_N]^T$ and compute a discrete distribution $\mathbf{p}^m = [p_1^m, ..., p_N^m]^T$, where $d_i \in [d_{\texttt{min}}, d_{\texttt{max}}]$ and $p_i^m$ is the probability that the ground truth disparity is $d_i$. The $\mathbf{p}^m$ is required to satisfy the conditions $p_i^m \geq 0$ and $\sum_i p_i^m = 1$.

The discrete formulation reasons the probability only at a finite set of disparities. However, in real-world applications, the ground-truth disparity is continuous. Therefore we propose to interpolate the discrete distribution by the Laplacian kernel, and the probability density function of disparity $x \in \mathbb{R}$ is computed by

$$p(x; \mathbf{p}^m) = \sum_i^N k(x, d_i) p_i^m \tag{1}$$

where $k(x, d_i)$ is defined as $\frac{1}{2\sigma} \exp -\frac{|x - d_i|}{\sigma}$, and $\sigma$ is the hyper-parameter for bandwidth. The above density function is valid because $p(x; \mathbf{p}^m) \geq 0$ for $\forall x \in \mathbb{R}$ and $\int p(x; \mathbf{p}^m) dx = 1$. An illustration of the interpolation is shown in Fig. 2 (c). The orange bars represent the given discrete distribution $\mathbf{p}^m$, and the green curve is the interpolated density function. In the following we show the continuous formulation enables us to compute the derivative of the risk function.

### 3.2 Risk of Disparity

To choose a value as the final prediction, we propose to minimize the following risk:

$$\texttt{argmin}_y F(y, \mathbf{p}^m) = \texttt{argmin}_y \int \mathcal{L}(y, x) p(x; \mathbf{p}^m) dx \tag{2}$$

where $F(y, \mathbf{p}^m)$ is called as the risk at $y$, and $\mathcal{L}(y, x)$ is the error function between $y$ and $x$. By risk we mean that if we take $y$ as predicted disparity, how much error there shall be with respect to the ground truth. Since the exact ground truth is unavailable at the time of making the prediction, we average the error across all possible ground-truth disparities with the distribution $p(x; \mathbf{p}^m)$.

Previous methods usually compute the expectation value of $x$ as the final prediction for the disparity:

$$y = \int x p(x; \mathbf{p}^m) dx. \tag{3}$$

In our framework, we can derive the same prediction when using the squared $L^2$ norm as the error function. More specifically, $\texttt{argmin}_y F(y, \mathbf{p}^m) = \int x p(x; \mathbf{p}^m) dx$ when $\mathcal{L}(y, x) = (y - x)^2$.

However, it is well known that the $L^2$ norm is not robust, and prones to outliers. As shown in Fig. 2 (b), the expectation is inaccurate when there are multiple modes in the distribution. Instead, we select the $L^1$ norm in our risk function:

$$\texttt{argmin}_y F(y, \mathbf{p}^m) = \texttt{argmin}_y \int |y - x| p(x; \mathbf{p}^m) dx. \tag{4}$$

Given the distribution $p(x; \mathbf{p}^m)$ of the disparity, the optimal $y$ will minimize the $L^1$ error with respect to all possible disparities weighted by the corresponding probability density. As shown in Fig. 2 (c), our final prediction is more robust to the incorrect modes and closer to the ground truth.

### 3.3 Differentiable Risk Minimization

One challenge of the $L^1$ norm is that there is no closed-form solution to the minimal risk in Eq.(4). To search for the optimal solution and enable end-to-end training, we introduce the details for the forward prediction and backward propagation below.

**Forward Prediction.** Given the discrete distribution $\mathbf{p}^m$, we find the optimal $y$ of Eq.(4) efficiently based on the following two observations. Firstly, the target function $F(y, \mathbf{p}^m)$ is convex with respect to $y$, thereby we find the optimal solution at where $\partial F/\partial y = 0$.

$$G(y, \mathbf{p}^m) \triangleq \frac{\partial F(y, \mathbf{p}^m)}{\partial y} = \sum_i p_i^m \texttt{Sign}(y - d_i)(1 - \exp -\frac{|y - d_i|}{\sigma}) = 0 \tag{5}$$

where $\texttt{Sign}()$ is the sign function, which a slight abuse of notation. $\texttt{Sign}()$ can be thought of as an indicator function, where it is 1 if $y > d_i$ and $-1$ otherwise. Secondly, the second-order derivative $\partial^2 F/\partial^2 y \geq 0$, so the first-order derivative is a non-decreasing function. We find the optimal disparity, i.e. the zero point of $G(y, \mathbf{p}^m)$, by binary search, as shown in Alg. 1 in the Appendix. In all experiments, we set the $\sigma$ and $\tau$ as 1.1 and 0.1 respectively. For $N$ disparity hypotheses, the binary search algorithm can find the optimal solution with time complexity of $O(\log N)$ (Cormen et al., 2009).

**Backward Propagation.** As alluded to above, the procedure of the forward prediction (Alg. 1) to solve Eq.(4) contains non-differentiable operations. However, to enable end-to-end training, we have to compute $dy/d\mathbf{p}^m$ to backward propagate the gradient. Our method is inspired by the Implicit Function Theorem (Krantz & Parks, 2002). More specifically, because $G(y, \mathbf{p}^m) \equiv 0$ at the optimal $y$, we obtain

$$dG(y, \mathbf{p}^m) = \frac{\partial G}{\partial y} dy + \frac{\partial G}{\partial \mathbf{p}^m} d\mathbf{p}^m = 0. \tag{6}$$

By organizing the terms, we obtain

$$\frac{dy}{d\mathbf{p}^m} = -\frac{\partial G/\partial \mathbf{p}^m}{\partial G/\partial y} = [\dots, \frac{\sigma \texttt{Sign}(d_i - y)(1 - \exp -\frac{|y - d_i|}{\sigma})}{\sum_j p_j^m \exp -\frac{|y - d_j|}{\sigma}}, \dots]^T. \tag{7}$$

We clip the denominator, i.e. $\sum_j p_j^m \exp -\frac{|y - d_j|}{\sigma}$ in the above equation to be no less than 0.1 to avoid large gradients.

### 3.4 Network Architecture

To find the disparity value, we match the image patches of left and right images by constructing stereo cost volumes, as in Kendall et al. (2017b) and Chang & Chen (2018). However, an exhaustive matching requires extensive memory and computation. For efficiency, we adopt a cascade structure following Gu et al. (2020). Specifically, we first sample the disparity hypothesis by a coarse matching, which is performed on low-resolution image features. The sampled hypothesis reduce the search space for matching to a large extent. Then we refine the sampled hypothesis at high-resolution image features. The overall pipeline is shown in Fig. 3, and includes 5 parts: (a) feature extraction (b) disparity hypotheses sampling (c) matching (d) cost aggregation (e) risk minimization. We introduce the details of each part below. More details are provided in the Appendix.

*(a)* **Feature Extraction.** Given an input image, the module aims to output multi-scale 2D feature maps. More specifically, we first use a ResNet (He et al., 2016) to extract 2D feature maps of resolution 1/4 and 1/2 with respect to the input image. The ResNet contains 4 stages of transformation with 3, 16, 3, 3 residual blocks respectively. And the spatial resolution is downsampled before the beginning of the first and third stages of transformation. Then we apply the spatial pyramid pooling (Zhao et al., 2017) on the 1/4-resolution feature map from the fourth stage to enlarge the receptive field. In the end, we upsample the enhanced feature map from 1/4 to 1/2 and fuse it with the 1/2-resolution feature map from ResNet. The final outputs are the feature maps of 1/4 and 1/2 resolution. We apply the same network and weights to extract features from left and right images.

*(b)* **Disparity Hypotheses Sampling.** The disparity hypotheses provide the candidates of pixel pairs to match. In the coarse stage, we sample 192 hypotheses uniformly within the range from 0 to the maximum possible disparity. In the refined stage, we reduce the sampling space according to the

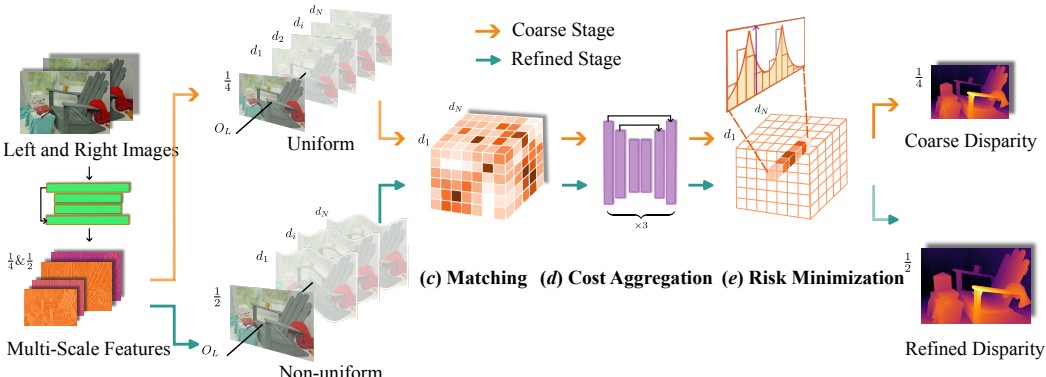

(a) Feature Extraction    (b) Disparity Hypotheses Sampling

Figure 3: **Overall pipeline (Left to Right).** We first extract multi-scale features from left and right images respectively. The subsequent procedures are divided into two stages. In the coarse stage —shown in orange arrow, we sample disparity hypotheses uniformly and match on 1/4-resolution features. While in the refined stage—shown in green arrow, to match 1/2-resolution features efficiently. Disparity hypotheses are sampled centering around the disparity predicted from the coarse stage. In both stages, we first construct cost volumes by concatenation, and then apply the stacked hourglass networks to aggregate the matching cost, and finally search for the disparity that minimizes the proposed $L^1$ risk in Eq.(4).

predicted disparity from the coarse stage. Specifically, for each pixel we sample 16 hypotheses between the maximum and minimum disparity in the local window of size $12 \times 12$.

*(c)* **Matching.** We match the 2D feature maps from the left and right images according to the sampled disparity hypothesis. The features at each pair of candidates pixels for matching will be concatenated along the channel dimension, which forms a 4D stereo cost volume (feature× disparity×height×width). In the coarse stage, we match the feature map of 1/4 resolution for efficiency. To capture high-frequency details, we match the 1/2-resolution feature map in the refined stage.

*(d)* **Cost Aggregation.** We use the stacked hourglass architecture (Newell et al., 2016) to transform the stereo cost volume and aggregate the matching cost. For the coarse and refined stages, the structures are the same except for the number of feature channels. Specifically, the network consists of three 3D hourglasss as in Chang & Chen (2018). Each hourglass first downsamples the volume hierarchically to 1/2 and 1/4 resolution with respect to the input volume, and then upsample in sequence to recover the resolution. The procedure helps aggregate information across various scales. The final output is a volume that represents the discrete distribution of disparity hypotheses.

*(e)* **Risk Minimization.** The module applies Alg. 1 to compute the optimal continuous disparity for each pixel given the discrete distribution of disparity hypotheses. During training, we additionally compute the gradient according to Eq.(7) to enable backward propagation.

### 3.5    LOSS FUNCTION

Given the predicted disparity $x^{\texttt{pred}} \in \mathbb{R}$ and the ground-truth disparity $x^{\texttt{gt}} \in \mathbb{R}$, we compute the smooth $L^1$ loss (Girshick, 2015):

$$\mathcal{L}(x^{\texttt{gt}}, x^{\texttt{pred}}) = \begin{cases} 0.5(x^{\texttt{gt}} - x^{\texttt{pred}})^2 & \text{if } |x^{\texttt{gt}} - x^{\texttt{pred}}| < 1.0 \\ |x^{\texttt{gt}} - x^{\texttt{pred}}| - 0.5 & \text{otherwise} \end{cases} \tag{8}$$

We apply the above loss function to the predicted disparities from both the coarse and refined stages, and obtain $\mathcal{L}_{\texttt{coarse}}$ and $\mathcal{L}_{\texttt{refined}}$ respectively. The total loss $\mathcal{L} = 0.1\mathcal{L}_{\texttt{coarse}} + 1.0\mathcal{L}_{\texttt{refined}}$.

## 4    EXPERIMENTS AND RESULTS

**Implementation Details.** We implement our method in PyTorch 2.0.1 (Python 3.11.2) with CUDA 11.8. The software is evaluated on a computing machine with GeForce-RTX-3090 GPU.

Table 1: Comparison with state-of-the-art methods on SceneFlow test set. The first and second bests are in red and blue respectively. **Our method** in bold.

| Method | Param (M) | Time (s) | EPE ↓ | > 0.5px ↓ | > 1px ↓ | > 2px ↓ |
|---|---|---|---|---|---|---|
| CFNet (Shen et al., 2021) | 21.98 | 0.13 | 1.04 | 15.91 | 10.30 | 6.89 |
| PCWNet (Shen et al., 2022) | 34.27 | 0.25 | 0.90 | 17.59 | 8.08 | 4.57 |
| ACVNet (Xu et al., 2022) | 6.84 | 0.16 | 0.47 | 9.70 | 5.00 | 2.74 |
| DLNR (Zhao et al., 2023) | 54.72 | 0.44 | 0.53 | 8.75 | 5.44 | 3.44 |
| IGEV (Xu et al., 2023) | 12.60 | 0.36 | 0.47 | 8.51 | 5.21 | 3.26 |
| **Ours** | **11.96** | **0.35** | **0.43** | **8.10** | **4.22** | **2.34** |

Table 2: Comparison with state-of-the-art methods on KITTI 2012 Benchmark. † denotes using extra data for pre-training. The first and second bests are in red and blue respectively. **Our method** in bold. The results are obtained from KITTI official website.

| Method | Param (M) | Time (s) | > 2px | | > 3px | |
|---|---|---|---|---|---|---|
| | | | Noc | All | Noc | All |
| LEAStereo (Cheng et al., 2020) | 1.81 | | 1.90 | 2.39 | 1.13 | 1.45 |
| CFNet (Shen et al., 2021) | 21.98 | 0.12 | 1.90 | 2.43 | 1.23 | 1.58 |
| ACVNet (Xu et al., 2022) | 6.84 | 0.15 | 1.83 | 2.34 | 1.13 | 1.47 |
| ACFNet (Chen et al., 2021) | | | 1.83 | 2.35 | 1.17 | 1.54 |
| NLCA-Net v2 (Rao et al., 2022) | | | 1.83 | 2.34 | 1.11 | 1.46 |
| CAL-Net (Chen et al., 2021) | | | 1.74 | 2.24 | 1.19 | 1.53 |
| CREStereo (Li et al., 2022) † | | | 1.72 | 2.18 | 1.14 | 1.46 |
| LaC+GANet (Liu et al., 2022a) | 9.43 | | 1.72 | 2.26 | 1.05 | 1.42 |
| IGEV (Xu et al., 2023)† | 12.60 | 0.32 | 1.71 | 2.17 | 1.12 | 1.44 |
| PCWNet (Shen et al., 2022) | 34.27 | 0.23 | 1.69 | 2.18 | 1.04 | 1.37 |
| **Ours** | **11.96** | **0.32** | **1.58** | **2.20** | **1.00** | **1.44** |

**Datasets.** We perform experiments on four datasets namely SceneFlow (Mayer et al., 2016), KITTI 2012 & 2015 (Geiger et al., 2012; Menze & Geiger, 2015), Middlebury 2014 (Scharstein & Szeliski, 2002), and ETH 3D (Schöps et al., 2017). **(a) SceneFlow** is a synthetic dataset containing 35,454 image pairs for training, and 4,370 image pairs for test. **(b) KITTI 2012 & 2015** are captured for autonomous driving. There are 194 training image pairs and 195 test image pairs in KITTI 2012. And there are 200 training image pairs and 200 test image pairs in KITTI 2015. **(c) Middlebury 2014** is an indoor dataset including 15 image pairs for training. **(d) ETH 3D** is a gray-scale dataset providing 27 image pairs for training.

**Training Details.** We train our network on SceneFlow. The weight is initialized randomly. We use AdamW optimizer (Loshchilov & Hutter, 2019) with weight decay $10^{-5}$. The learning rate decreases from $2 \times 10^{-4}$ to $2 \times 10^{-8}$ according to the one cycle learning rate policy. We train the network for $2 \times 10^5$ iterations. The images will be randomly cropped to $320 \times 736$. For KITTI 2012 & 2015 benchmarks, we further fine tune the network on the training image pairs for $2.5 \times 10^3$ iterations. The learning rate starts from $5 \times 10^{-5}$ to $5 \times 10^{-9}$. More details are provided in the Appendix.

## 4.1 IN-DOMAIN EVALUATION

Tab.(1), Tab.(2) and Tab.(3) provide statistical comparison results with the competing methods on SceneFlow, KITTI 2012 & 2015 bechmarks, respectively. All the methods have been trained or fine-tuned on the corresponding training set. In SceneFlow test set, our proposed approach shows the best results for all the evaluation metrics. Particularly, we reduce the > 1px error from 5.00 to 4.22, and the > 0.5px error from 8.51 to 8.10. In KITTI 2012 & 2015 benchmarks, the matching accuracy of our approach in the non-occluded regions rank the first among the published methods. Especially, in KITTI 2012, we reduce the > 2px error in non-occluded regions by 0.11.

## 4.2 CROSS-DOMAIN GENERALIZATION

In this part, we compare the methods when dealing with environments never seen in the training set. Specifically, all methods are trained only on SceneFlow training set, and then evaluated on the training set of Middlebury, ETH 3D and KITTI 2012 & 2015 *without* fine-tuning.

The statistical comparison results are shown in Tab.(4), Tab.(5), Tab.(6) and Tab.(7), respectively. Our proposed approach achieves the first or the second best accuracies under all the evaluation metrics on the four real-world datasets. Particularly, for Middlebury we reduce the > 1px error

Table 3: Comparison with state-of-the-art methods on KITTI 2015 Benchmark. † denotes using extra data for pre-training. The first and second bests are in red and blue respectively. **Our method** in bold. The results are obtained from KITTI official website.

| Method | Param (M) | Time (s) | All | | | Noc | | |
|---|---|---|---|---|---|---|---|---|
| | | | D1_bg | D1_fg | D1_all | D1_bg | D1_fg | D1_all |
| LEAStereo (Cheng et al., 2020) | 1.81 | | 1.40 | 2.91 | 1.65 | 1.29 | 2.65 | 1.51 |
| CFNet (Shen et al., 2021) | 21.98 | 0.12 | 1.54 | 3.56 | 1.88 | 1.43 | 3.25 | 1.73 |
| ACVNet (Xu et al., 2022) | 6.84 | 0.15 | 1.37 | 3.07 | 1.65 | 1.26 | 2.84 | 1.52 |
| ACFNet (Chen et al., 2021) | | | 1.51 | 3.80 | 1.89 | 1.36 | 3.49 | 1.72 |
| NLCA-Net v2 (Rao et al., 2022) | | | 1.41 | 3.56 | 1.77 | 1.28 | 3.22 | 1.60 |
| CAL-Net (Chen et al., 2021) | | | 1.59 | 3.76 | 1.95 | 1.45 | 3.42 | 1.77 |
| CREStereo (Li et al., 2022) † | | | 1.45 | 2.86 | 1.69 | 1.33 | 2.60 | 1.54 |
| LaC+GANet (Liu et al., 2022a) | 9.43 | | 1.44 | 2.83 | 1.67 | 1.26 | 2.64 | 1.49 |
| IGEV (Xu et al., 2023) † | 12.60 | 0.32 | 1.38 | 2.67 | 1.59 | 1.27 | 2.62 | 1.49 |
| DLNR (Zhao et al., 2023) | 54.72 | 0.39 | 1.60 | 2.59 | 1.76 | 1.45 | 2.39 | 1.61 |
| PCWNet (Shen et al., 2022) | 34.27 | 0.23 | 1.37 | 3.16 | 1.67 | 1.26 | 2.93 | 1.53 |
| CroCo-Stereo (Weinzaepfel et al., 2023)† | 417.15 | | 1.38 | 2.65 | 1.59 | 1.30 | 2.56 | 1.51 |
| **Ours** | **11.96** | **0.32** | **1.40** | **2.76** | **1.63** | **1.25** | **2.62** | **1.48** |

Table 4: Cross-domain evaluation on Middlebury training set of quarter resolution. † denotes using extra data for pre-training. The first and second bests are in red and blue respectively. **Our method** in bold. All methods are trained on SceneFlow and evaluated on Middlebury training set without fine-tuning.

| Method | Param (M) | Time (s) | > 0.5px | | > 1px | |
|---|---|---|---|---|---|---|
| | | | Noc | All | Noc | All |
| CFNet (Shen et al., 2021) | 21.98 | 0.11 | 29.50 | 34.30 | 17.85 | 22.16 |
| ACVNet (Xu et al., 2022) | 6.84 | 0.12 | 39.04 | 42.97 | 22.68 | 26.49 |
| DLNR (Zhao et al., 2023) | 12.60 | 0.63 | 19.43 | 23.75 | 10.16 | 13.76 |
| IGEV (Xu et al., 2023)† | 12.60 | 0.34 | 19.05 | 23.33 | 10.44 | 14.05 |
| PCWNet (Shen et al., 2022) | 34.27 | 0.19 | 33.33 | 38.00 | 16.80 | 21.36 |
| **Ours** | **11.96** | **0.25** | **19.22** | **23.33** | **9.32** | **12.63** |

from 13.76 to 12.63. Further more, on ETH 3D we reduce the $> 0.5$px error from 10.39 to 8.59, and $> 1$px error from 4.05 to 2.71. It can be observed our approach is more robust and generalizes better than recent state of the arts on the cross-domain setting.

The qualitative comparison is presented in the Appendix.

## 4.3 ABLATION STUDIES

In this subsection, we perform ablation studies to analyze the effects of the risk minimization method for disparity prediction. All the models are trained on SceneFlow and then tested on Middlebury *without* fine-tuning.

*(a)* **Effect of Risk Minimization.** We compare the expectation and the $L^1$-norm risk minimization for disparity prediction during training and test. We present the comparison results in Tab.(8). Even using the expectation to predict disparities during training, we still slightly improve the accuracy by changing to the $L^1$-norm risk minimization during test. Moreover, if we use the $L^1$-norm risk minimization in both training and test, the best accuracy is achieved under all metrics.

*(b)* **Performance with Different Networks.** We replace the disparity prediction method in ACVNet (Xu et al., 2022) and PCWNet (Shen et al., 2022) from expectation to $L^1$-norm risk minimization

Table 5: Cross-domain evaluation on ETH 3D training set. † denotes using extra data for pre-training. The first and second bests are in red and blue respectively. **Our method** in bold. All methods are trained on SceneFlow and evaluated on ETH 3D training set without fine-tuning.

| Method | Param (M) | Time (s) | > 0.5px | | > 1px | |
|---|---|---|---|---|---|---|
| | | | Noc | All | Noc | All |
| CFNet (Shen et al., 2021) | 21.98 | 0.11 | 15.57 | 16.24 | 5.30 | 5.59 |
| ACVNet (Xu et al., 2022) | 6.84 | 0.12 | 21.83 | 22.64 | 8.13 | 8.81 |
| DLNR (Zhao et al., 2023) | 12.60 | 0.34 | 18.66 | 19.07 | 13.11 | 13.39 |
| IGEV (Xu et al., 2023)† | 12.60 | 0.29 | 9.83 | 10.39 | 3.60 | 4.05 |
| PCWNet (Shen et al., 2022) | 34.27 | 0.20 | 18.25 | 18.88 | 5.17 | 5.43 |
| **Ours** | **11.96** | **0.26** | **7.90** | **8.59** | **2.41** | **2.71** |

Table 6: Cross-domain evaluation on KITTI 2012 training set. † denotes using extra data for pre-training. The first and second bests are in red and blue respectively. **Our method** in bold. All methods are trained on SceneFlow and evaluated on KITTI 2012 training set without fine-tuning.

| Method | Param (M) | Time (s) | > 2px | | > 3px | |
| --- | --- | --- | --- | --- | --- | --- |
| | | | Noc | All | Noc | All |
| CFNet (Shen et al., 2021) | 21.98 | 0.12 | 7.08 | 7.97 | 4.66 | 5.31 |
| ACVNet (Xu et al., 2022) | 6.84 | 0.15 | 20.34 | 21.44 | 14.22 | 15.18 |
| DLNR (Zhao et al., 2023) | 12.60 | 0.39 | 12.01 | 12.81 | 8.83 | 9.46 |
| IGEV (Xu et al., 2023)† | 12.60 | 0.32 | 7.55 | 8.44 | 5.03 | 5.70 |
| PCWNet (Shen et al., 2022) | 34.27 | 0.23 | 6.63 | 7.49 | 4.08 | 4.68 |
| **Ours** | **11.96** | **0.32** | **5.82** | **6.70** | **3.84** | **4.43** |

Table 7: Cross-domain evaluation on KITTI 2015 training set. † denotes using extra data for pre-training. The first and second bests are in red and blue respectively. **Our method** in bold. All methods are trained on SceneFlow and evaluated on KITTI 2015 training set without fine-tuning.

| Method | Param (M) | Time (s) | All | | | Noc | | |
| --- | --- | --- | --- | --- | --- | --- | --- | --- |
| | | | D1_bg | D1_fg | D1_all | D1_bg | D1_fg | D1_all |
| CFNet (Shen et al., 2021) | 21.98 | 0.12 | 4.77 | 13.26 | 6.07 | 4.64 | 12.88 | 5.88 |
| ACVNet (Xu et al., 2022) | 6.84 | 0.15 | 12.35 | 19.97 | 13.52 | 12.04 | 18.82 | 13.06 |
| DLNR (Zhao et al., 2023) | 9.43 | 0.39 | 18.67 | 14.86 | 18.08 | 18.42 | 14.18 | 17.78 |
| IGEV (Xu et al., 2023) † | 12.60 | 0.32 | 4.01 | 15.58 | 5.79 | 3.88 | 14.94 | 5.55 |
| PCWNet (Shen et al., 2022) | 34.27 | 0.23 | 4.25 | 14.40 | 5.81 | 4.11 | 13.95 | 5.60 |
| **Ours** | **11.96** | **0.32** | **3.68** | **13.52** | **5.19** | **3.57** | **13.05** | **5.00** |

*only* during test. The results are shown in Tab.(8). Our proposed method improves the accuracy under all metrics *without* re-training.

## 4.4 NETWORK PROCESSING TIME & PAREMETERS

We present the networks' inference time and number of parameters in Tab.(1), Tab.(2), Tab.(4), and Tab.(5). For a fair comparison, all networks are evalutated on the same machine with a GeForce-RTX-3090 GPU. Our network outperforms many state of the arts on inference time, including IGEV and DLNR. Moreover, our network has fewer learnable parameters than PCWNet, IGEV and DLNR.

In addition, our proposed $L^1$-norm risk minimization module doesn't require extra learnable parameters. The running time is shown in Tab.(8). By changing the disparity prediction method from expectation to our proposed approach, the running time is slightly increased.

Table 8: Ablation studies on Middlebury training set of quarter resolution. The first and second bests are in red and blue respectively. **Our method** in bold. All methods are trained on SceneFlow and evaluated on Middlebury training set without fine-tuning.

| Backbone | Training | Test | Param (M) | Time (s) | > 1px | | > 2px | |
| --- | --- | --- | --- | --- | --- | --- | --- | --- |
| | | | | | Noc | All | Noc | All |
| ACVNet(Xu et al., 2022) | Expectation | Expectation | 6.84 | 0.12 | 22.68 | 26.49 | 13.54 | 16.49 |
| | Expectation | L1-Risk | 6.84 | 0.18 | 22.32 | 26.14 | 13.13 | 16.05 |
| PCWNet(Shen et al., 2022) | Expectation | Expectation | 34.27 | 0.19 | 16.80 | 21.36 | 8.93 | 12.62 |
| | Expectation | L1-Risk | 34.27 | 0.26 | 16.53 | 21.08 | 8.65 | 12.30 |
| **Ours** | Expectation | Expectation | 11.96 | 0.17 | 9.88 | 13.27 | 4.92 | 7.29 |
| | Expectation | L1-Risk | 11.96 | 0.25 | 9.83 | 13.22 | 4.90 | 7.27 |
| | L1-Risk | Expectation | 11.96 | 0.17 | 9.83 | 13.19 | 4.79 | 7.06 |
| | **L1-Risk** | **L1-Risk** | **11.96** | **0.25** | **9.32** | **12.63** | **4.49** | **6.70** |

## 4.5 CONCLUSION

Our work provides a novel way of thinking and solving stereo-matching problems in computer vision via the principle of risk minimization (Vapnik, 1991). The paper provides in-depth theoretical and practical benefits of using our proposed formulation. It is shown that the presented approach is more robust to multi-modal distributions and outliers, and generalizes better on cross-domain stereo images. Furthermore, a new mathematical fabric to research stereo-matching problems is presented, enabling adaptations from fields such as robotics and control engineering.

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

## A  APPENDIX

### A.1  TRAINING DETAILS

We train our network on SceneFlow. The weight is initialized randomly. We use AdamW optimizer (Loshchilov & Hutter, 2019) with weight decay $10^{-5}$. The learning rate decreases from $2 \times 10^{-4}$ to $2 \times 10^{-8}$ according to the one cycle learning rate policy. We train the network for $2 \times 10^5$ iterations. The images will be randomly cropped to $320 \times 736$. For KITTI 2012 & 2015 benchmarks, we further fine tune the network on the training image pairs for $2.5 \times 10^3$ iterations. The learning rate starts from $5 \times 10^{-5}$ to $5 \times 10^{-9}$.

Following RAFT-Stereo (Lipson et al., 2021), we apply various image augmentations during training to avoid the over-fitting problem. Specifically, the augmentations include *(a)* color transformation, *(b)* occlusion, and *(c)* spatial transformation. In *(a)* color transformation, we randomly change the

Table 9: Network structure for feature extraction.

| Name | Layer Setting | | Output Dimension |
|---|---|---|---|
| | ResNet | | |
| Input | | | $H \times W \times 3$ |
| Stem-1 | $3 \times 3, 32$ | | $H \times W \times 32$ |
| Stem-2 | $3 \times 3, 32$ | | $H \times W \times 32$ |
| Stem-3 | $3 \times 3, 32$ | | $\frac{1}{2}H \times \frac{1}{2}W \times 32$ |
| Stage-1 | $\begin{array}{c} 3 \times 3, 32 \\ 3 \times 3, 32 \end{array}$ | $\times 3$ | $\frac{1}{2}H \times \frac{1}{2}W \times 32$ |
| Stage-2 | $\begin{array}{c} 3 \times 3, 64 \\ 3 \times 3, 64 \end{array}$ | $\times 16$ | $\frac{1}{4}H \times \frac{1}{4}W \times 64$ |
| Stage-3 | $\begin{array}{c} 3 \times 3, 128 \\ 3 \times 3, 128 \end{array}$ | $\times 3$ | $\frac{1}{4}H \times \frac{1}{4}W \times 128$ |
| Stage-4 | $\begin{array}{c} 3 \times 3, 128 \\ 3 \times 3, 128 \end{array}$ | $\times 3, \texttt{dila} = 2$ | $\frac{1}{4}H \times \frac{1}{4}W \times 128$ |
| | Spatial Pyramid Pooling | | |
| Branch-1 | $64 \times 64$ avg pool
$3 \times 3, 32$
bilinear interpolation | | $\frac{1}{4}H \times \frac{1}{4}W \times 32$ |
| Branch-2 | $32 \times 32$ avg pool
$3 \times 3, 32$
bilinear interpolation | | $\frac{1}{4}H \times \frac{1}{4}W \times 32$ |
| Branch-3 | $16 \times 16$ avg pool
$3 \times 3, 32$
bilinear interpolation | | $\frac{1}{4}H \times \frac{1}{4}W \times 32$ |
| Branch-4 | $8 \times 8$ avg pool
$3 \times 3, 32$
bilinear interpolation | | $\frac{1}{4}H \times \frac{1}{4}W \times 32$ |
| Concat [Stage-2, Stage-4, Branch-1, Branch-2, Branch-3, Branch-4] | | | $\frac{1}{4}H \times \frac{1}{4}W \times 32$ |
| Fusion-1 | $\begin{array}{c} 3 \times 3, 128 \\ 1 \times 1, 32 \end{array}$ | | $\frac{1}{4}H \times \frac{1}{4}W \times 32$ |
| | UpSample | | |
| Up-1 | nearest interpolation | | $\frac{1}{2}H \times \frac{1}{2}W \times 32$ |
| Add [Stage-1, Up-0] | | | $\frac{1}{2}H \times \frac{1}{2}W \times 32$ |
| Fusion-2 | $3 \times 3, 16$ | | $\frac{1}{2}H \times \frac{1}{2}W \times 16$ |

brightness, contrast, saturation and hue of the left and right images independently. The brightness and contrast factors are uniformly chosen from [0.6, 1.4]. The saturation factor is uniformly chosen from [0.0, 1.4]. The hue factor is uniformly chosen from [-0.16, 0.16]. In *(b)* occlusion, we randomly select a few rectangular regions in the right image, and set the pixels inside the regions as the mean color of the right image. The number of regions is chosen from {0, 1, 2, 3} with probabilities {0.5, 0.166, 0.166, 0.166}. The position of the region is uniformly chosen in the right image, and the width and height are uniformly chosen from [50, 100]. In *(c)* spatial transformation, we randomly crop the left and right images to the resolution 320×736.

## A.2 NETWORK STRUCTURE DETAILS

In this part, we present more details for the *(i)* feature extraction and *(ii)* cost aggregation.

*(i)* **Feature Extraction.** Given an input image, the module aims to output multi-scale 2D feature maps. More specifically, we first use a ResNet (He et al., 2016) to extract 2D feature maps of resolution 1/4 and 1/2 with respect to the input image. The ResNet contains 4 stages of non-linear transformation with 3, 16, 3, 3 residual blocks respectively, where each block is composed of convolutional layers and skip connections. And the spatial resolution is downsampled before the beginning of the first and third stages of transformation. Then we apply the spatial pyramid pooling (Zhao et al., 2017) on the 1/4-resolution feature map from the fourth stage of transformation to enlarge the receptive field. In the end, we upsample the enhanced feature map from 1/4 to 1/2 and fuse it with the 1/2-resolution feature map from the first stage of transformation in ResNet. The final outputs are the feature maps of 1/4 and 1/2 resolution. We apply the same network and weights to extract features from left and right images. The details of the network structure and the resolution of the feature maps are shown in Tab.(9).

*(ii)* **Cost Aggregation.** We use the stacked hourglass architecture (Newell et al., 2016) to transform the stereo cost volume and aggregate the matching cost. For the coarse and refined stages, the structures are the same except for the number of feature channels. Specifically, the network consists of three 3D hourglasss as in Chang & Chen (2018). Each hourglass first downsamples the volume

Table 10: Network structure for 3D hourglass.

| Name | Layer Setting | Output Dimension |
|---|---|---|
| Input | | $D \times H \times W \times C$ |
| Conv-1 | $3 \times 3 \times 3, 2C$ | $\frac{1}{2}D \times \frac{1}{2}H \times \frac{1}{2}W \times 2C$ |
| Conv-2 | $3 \times 3 \times 3, 2C$ | $\frac{1}{2}D \times \frac{1}{2}H \times \frac{1}{2}W \times 2C$ |
| Conv-3 | $3 \times 3 \times 3, 4C$ | $\frac{1}{4}D \times \frac{1}{4}H \times \frac{1}{4}W \times 4C$ |
| Conv-4 | $3 \times 3 \times 3, 4C$ | $\frac{1}{4}D \times \frac{1}{4}H \times \frac{1}{4}W \times 4C$ |
| Atte-4 | $3 \times 3 \times 3, C$ 
 $3 \times 3 \times 3, 4C$ 
 `sigmoid` 
 `prod` Conv-4 | $\frac{1}{4}D \times \frac{1}{4}H \times \frac{1}{4}W \times 4C$ |
| Conv-5 | `deconv` $3 \times 3 \times 3, 2C$ 
 `add` Conv-2 | $\frac{1}{2}D \times \frac{1}{2}H \times \frac{1}{2}W \times 2C$ |
| Atte-5 | $3 \times 3 \times 3, C$ 
 $3 \times 3 \times 3, 2C$ 
 `sigmoid` 
 `prod` Conv-5 | $\frac{1}{2}D \times \frac{1}{2}H \times \frac{1}{2}W \times 2C$ |
| Conv-6 | `deconv` $3 \times 3 \times 3, C$ 
 `add` Input | $D \times H \times W \times C$ |
| Atte-6 | $3 \times 3 \times 3, C$ 
 $3 \times 3 \times 3, C$ 
 `sigmoid` 
 `prod` Conv-6 | $D \times H \times W \times C$ |

---

**Algorithm 1** Forward Prediction

---

**Require:** $\tau > 0$, $\sigma > 0$, $\mathbf{d} = [d_1, ..., d_N]$, $d_1 < d_2 < \cdots < d_N$, and $\mathbf{p}^m = [p_1^m, ..., p_N^m]$

$\quad d^l \leftarrow d_1$           ▷ Initialize search boundaries

$\quad d^r \leftarrow d_N$

$\quad g \leftarrow \tau + 1$           ▷ Initialize the derivative

$\quad$ **while** $|g| > \tau$ **do**

$\quad\quad d^m \leftarrow (d^l + d^r)/2.0$           ▷ Compute the mid point

$\quad\quad g \leftarrow \sum_i p_i^m \mathtt{Sign}(d^m - d_i)(1 - \exp{-\frac{|d^m - d_i|}{\sigma}})$    ▷ Compute the derivative by Eq.(5)

$\quad\quad$ **if** $g > 0$ **then**           ▷ Update search boundaries

$\quad\quad\quad d^r \leftarrow d^m$

$\quad\quad$ **else**

$\quad\quad\quad d^l \leftarrow d^m$

$\quad\quad$ **end if**

$\quad$ **end while**

$\quad$ **return** $d^m$           ▷ Return the mid point

---

hierarchically to 1/2 and 1/4 resolution with respect to the input volume, and then upsamples in sequence to recover the resolution. The above procedure helps aggregate the matching information across various scales. The final output is a volume that represents the discrete distribution of disparity hypotheses. We present the details of a single hourglass structure in Tab.(10). For an input image with resolution $h \times w$, the $D$, $H$, $W$, $C$ are 192, $h/4$, $w/4$, 32 respectively in the coarse stage. In the refined stage, we set $D$, $H$, $W$, $C$ to be 16, $h/2$, $w/2$, 16 respectively.

### A.3 FORWARD PREDICTION ALGORITHM

In this section, we introduce the details of the forward prediction. As shown in Alg. 1, we adopt the binary search algorithm to search for the optimal disparity for each pixel. The inputs to the algorithm include the tolerance $\tau$, the bandwidth $\sigma$ of the Laplacian kernel, the disparity hypotheses $\mathbf{d}$, and the discrete distribution $\mathbf{p}^m$. The output is the optimal disparity.

### A.4 EXPERIMENTS

#### A.4.1 ABLATION STUDY FOR TOLERANCE

In this part, we change the value of the tolerance $\tau$ in the binary search algorithm and observe its effects. As shown in Tab.(11), when decreasing the value of $\tau$, the search algorithm will iterate for

more times to search for the optimal solution. And the error of the predicted disparity is reduced. When $\tau \geq 0.1$, the algorithm achieves the best accuracy.

Table 11: Ablation studies for tolerance $\tau$ on Middlebury training set of quarter resolution. The first and second bests are in red and blue respectively. **Our method** in bold. All settings are trained on SceneFlow and evaluated on Middlebury training set without fine-tuning.

| Tolerance $\tau$ | Number of Iterations | > 1px | | > 2px | |
|---|---|---|---|---|---|
| | | Noc | All | Noc | All |
| 0.3 | 9 | 9.36 | 12.67 | 4.50 | 6.71 |
| **0.1** | **11** | **9.32** | **12.63** | **4.49** | **6.70** |
| 0.01 | 14 | 9.32 | 12.63 | 4.49 | 6.70 |

### A.4.2 ABLATION STUDIES FOR HUBER LOSS

In this part, we evaluate the effects of different loss functions. In Tab.(12), we evaluate the $L^2$ loss, the $L^1$ loss, and the Huber loss, i.e. a combination of $L^1$ and $L^2$ norm depending on the thresholding value $\beta$. The table clearly shows the benefit of using risk minimization loss under $L^1$.

Table 12: Ablation studies for loss function on Middlebury training set of quarter resolution. The first and second bests are in red and blue respectively. **Our method** in bold. All settings are trained on SceneFlow and evaluated on Middlebury training set without fine-tuning.

| Loss | > 1px | | > 2px | |
|---|---|---|---|---|
| | Noc | All | Noc | All |
| $L^2$ | 9.83 | 13.19 | 4.79 | 7.06 |
| $\beta = 10.0$ | 9.41 | 12.73 | 4.55 | 6.76 |
| $\beta = 4.0$ | 9.36 | 12.68 | 4.51 | 6.72 |
| $\beta = 1.0$ | 9.33 | 12.64 | 4.50 | 6.70 |
| $L^1$ | **9.32** | **12.63** | **4.49** | **6.70** |

### A.4.3 ABLATION STUDIES FOR NETWORK ARCHITECTURES

In this part, we apply our method to the IGEV (Xu et al., 2023) framework. Specifically, we use our method to compute the initial disparities from the geometry encoding volume. The results are shown in Tab.(13). Our method improves the accuracy of IGEV.

Table 13: Ablation studies for IGEV on Middlebury training set of quarter resolution. The first and second bests are in red and blue respectively. **Our method** in bold. All methods are trained on SceneFlow and evaluated on Middlebury training set without fine-tuning.

| Backbone | Training | Test | Param (M) | Time (s) | > 3px | | > 4px | |
|---|---|---|---|---|---|---|---|---|
| | | | | | Noc | All | Noc | All |
| IGEV(Xu et al., 2023) | Expectation | Expectation | 12.60 | 0.34 | 4.47 | 6.64 | 3.46 | 5.32 |
| | **Expectation** | **L1-Risk** | **12.60** | **0.38** | **4.37** | **6.63** | **3.40** | **5.32** |

### A.4.4 ABLATION STUDIES FOR INTERPOLATION KERNEL

In this part, we change the interpolation kernel from Laplacian to Gaussian and observe the effects. As shown in Tab.(14), we find the Laplacian kernel has better accuracy.

### A.4.5 CROSS-DOMAIN GENERALIZATION

In this part, we apply our method to ITSA (Chuah et al., 2022) only at inference time. We use the pre-trained model provided by ITSA, which is trained on synthetic images. As shown in Tab.(15), when evaluated on real-world datasets, our method can improve the performance on various networks and benchmarks.

Table 14: Ablation studies for interpolation kernel on Middlebury training set of quarter resolution. The first and second bests are in red and blue respectively. **Our method** in bold. All settings are trained on SceneFlow and evaluated on Middlebury training set without fine-tuning.

| Kernel | Param (M) | Time (s) | > 1px | | > 2px | |
|---|---|---|---|---|---|---|
| | | | Noc | All | Noc | All |
| Gaussian | 11.96 | 0.25 | 9.35 | 12.66 | 4.50 | 6.71 |
| **Laplacian** | **11.96** | **0.25** | **9.32** | **12.63** | **4.49** | **6.70** |

Table 15: Cross-domain evaluation with ITSA. The first and second bests are in red and blue respectively. All methods are trained on SceneFlow and evaluated on Middlebury training set without fine-tuning.

| Backbone | Training | Test | KITTI 2012 | KITTI 2015 | Middlebury | ETH3D |
|---|---|---|---|---|---|---|
| ITSA-PSMNet | Expectation | Expectation | 5.2 | 5.8 | 9.6 | 9.8 |
| | Expectation | L1-Risk | 5.0 | 5.6 | 9.0 | 9.7 |
| ITSA-GwcNet | Expectation | Expectation | 4.9 | 5.4 | 9.3 | 7.1 |
| | Expectation | L1-Risk | 4.6 | 5.2 | 8.8 | 7.1 |
| ITSA-CFNet | Expectation | Expectation | 4.2 | 4.7 | 8.5 | 5.1 |
| | Expectation | L1-Risk | 4.1 | 4.7 | 8.4 | 5.0 |

## A.5  QUALITATIVE RESULTS

In this section, we present more qualitative results on real-world datasets in Fig. 4, Fig. 5 and Fig. 6. It can be observed that in general our method generalizes and predicts high-frequency details better than other recent methods.

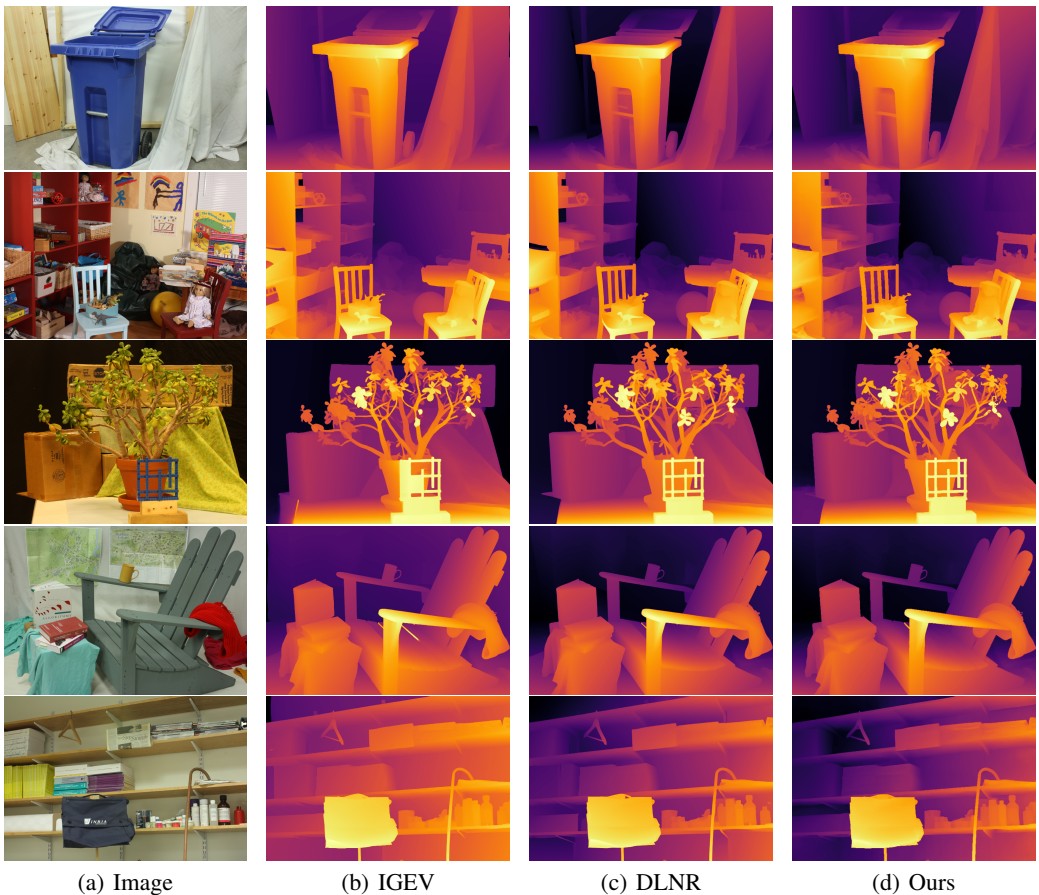

(a) Image      (b) IGEV      (c) DLNR      (d) Ours

Figure 4: **Qualitative Comparison.** We compare our method with recent state-of-the-art methods such as IGEV (Xu et al., 2023), DLNR (Zhao et al., 2023) on Middlebury (Scharstein & Szeliski, 2002). All methods are trained only on SceneFlow (Mayer et al., 2016), and evaluated at quarter resolution.

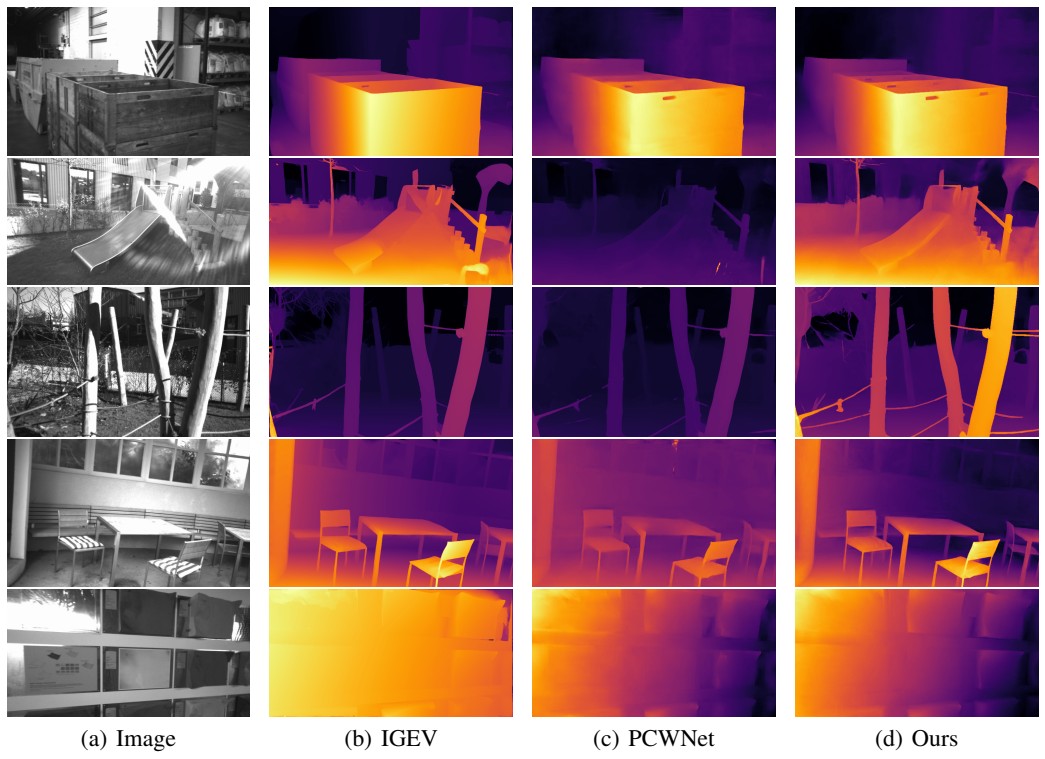

(a) Image      (b) IGEV      (c) PCWNet      (d) Ours

Figure 5: **Qualitative Comparison.** We compare our method with recent state-of-the-art methods such as IGEV (Xu et al., 2023), PCWNet (Shen et al., 2022) on ETH 3D (Schöps et al., 2017). All methods are trained only on SceneFlow (Mayer et al., 2016).

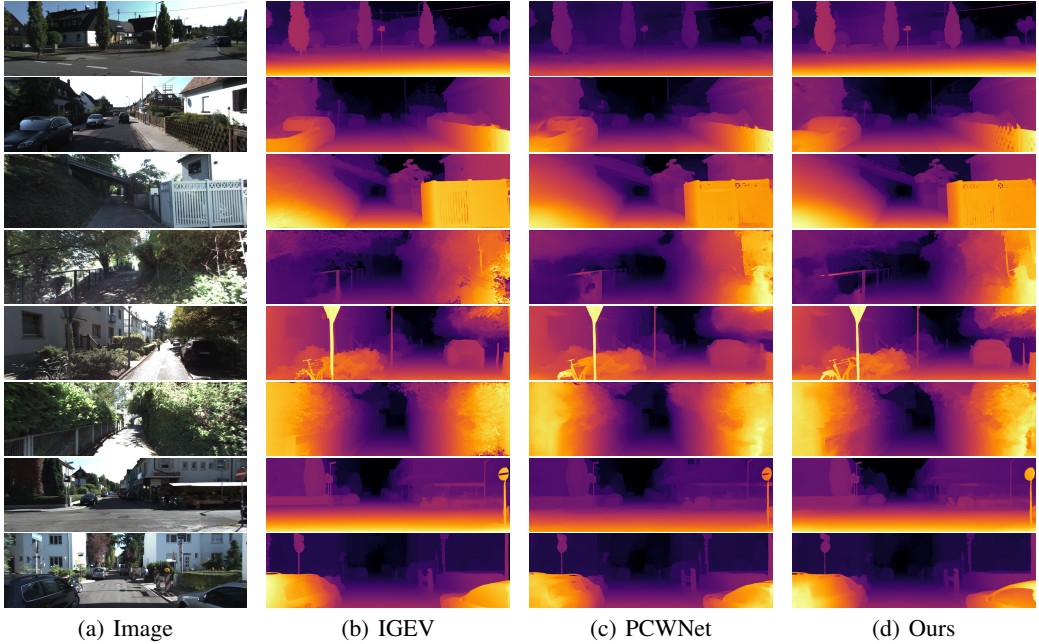

(a) Image      (b) IGEV      (c) PCWNet      (d) Ours

Figure 6: **Qualitative Comparison.** We compare our method with recent state-of-the-art methods such as IGEV (Xu et al., 2023), PCWNet (Shen et al., 2022) on KITTI 2012 (Geiger et al., 2012). All methods are trained only on SceneFlow (Mayer et al., 2016).

