# OpenReview forum: "Robust Stereo Matching by Risk Minimization"
_ICLR.cc/2024/Conference — Submitted to ICLR 2024_

### Official Review · Reviewer_a5XH · 2023-10-28

**Soundness:** 3 good
**Presentation:** 3 good
**Contribution:** 3 good
**Rating:** 6
**Confidence:** 5

**Summary:**

The authors present to capture continuous disparity in current stereo matching networks. Specifically, they propose to treat the disparity prediction problem as an optimal solution to the risk minimization problem. They explore the implicit function theorem and propose a differentiable scheme for end-to-end network training with risk minimization. Their method can improve the robustness and fine-grained prediction quality of stereo matching networks.

**Strengths:**

1. The paper introduces a fresh risk minimum perspective on the challenge of capturing continuous disparity for stereo matching.
2. One notable strength of this method is its potential to enhance the fine-grained prediction quality in stereo matching. The ability to capture continuous disparity offers a crucial advantage in achieving more accurate and detailed depth maps.
3. The method can be applied to existing stereo matching networks without re-training and improves their performance.
4. The paper is well-written and easy to understand.

**Weaknesses:**

1. The current iterative optimization based methods such as RAFT-Stereo directly predict continuous prediction and show strong robustness. It seems the method in this paper is not suitable for these iterative optimization methods, which I consider a major weakness restricting its application.
2. The baseline model seems to have strong robustness and the benefit of applying risk minimization is relatively small, which I consider as a minor weakness.

**Questions:**

1. How important is the Laplacian kernel for interpolation in Eq.(1)? I wonder if using the simple bilinear interpolation is enough for capturing continuous cost values.
2. Considering the paper builds its own network architecture, I wonder how the method can improve existing cost aggregation based networks such as PCWNet or GWCNet.

---

> ### Author Response · Authors · 2023-11-22
>
> ## Q1: The current iterative optimization based methods such as RAFT-Stereo directly predict continuous prediction and show strong robustness. It seems the method in this paper is not suitable for these iterative optimization methods, which I consider a major weakness restricting its application.
> A1: Thank you for highlighting this. This comment of yours aligns well with the advice provided by R2. Following this,  we performed experiments IGEV, an iterative optimization method. We have added Table 13 in the revised draft showing the benefit of our iterative optimization approach with a well-known existing  architecture in stereo matching.
>
> ## Q2: How important is the Laplacian kernel for interpolation in Eq.(1)? I wonder if using the simple bilinear interpolation is enough for capturing continuous cost values.
> A2: The importance of the Laplacian kernel can be observed from the Table 14 added in the revised draft. Using other popular kernel such as Gaussian kernel degrades the performance and over-smooths the solution, whereas, the Laplacian kernel preserves high-frequency details providing better results. We have added this ablation study on the choice of kernel in the appendix—refer Table 14.
>
> ## Q3: Considering the paper builds its own network architecture, I wonder how the method can improve existing cost aggregation based networks such as PCWNet or GWCNet.
> A3: In the revised draft, we have a table — Table 8, showing the benefit of our proposed optimization when applied to existing cost aggregation based networks such as PCWNet.

---

### Official Review · Reviewer_bKMt · 2023-10-30

**Soundness:** 3 good
**Presentation:** 4 excellent
**Contribution:** 3 good
**Rating:** 6
**Confidence:** 4

**Summary:**

This paper introduces a novel risk minimization based stereo matching method to capture the continuous disparity. While the state-of-the-art methods utilize expectation value of the distribution of multiple disparity candidates, the proposed method is searching the continuous disparity with the minimum risk or the smallest prediction error with the ground truth. In addition, it utilizes L1 formulation that shows more robust performance in the experimental results. To enable the end-to-end network training of L1 risk minimization, it introduces a differentiable scheme to forward and backward propagation. The proposed method is more robust to the regions with multiple disparity candidates and outliers. Experimental results show that the proposed method has better generalization in cross-domain stereo images.

**Strengths:**

+ Unlike conventional methods, the paper introduces a novel risk of disparity concept to measure the final disparity. The final disparity is predicted by minimizing the prediction error of all possible ground-truth disparities (Note that the exact ground truth is unavailable during the prediction). The paper shows that the minimizing the L2 risk is similar to the expectation value computation to predict final disparity, which is not robust to outlier. Thus, it proposes to utilize L1 norm which is more robust to predict final disparity with multiple disparity candidates or outliers.

+ The paper introduces a method to forward and backward propagation of a L1 risk minimization process. The final disparity is done by performing binary search across disparity candidates by selecting the disparity with minimum risk. To effectively backward propagate the gradient, the method is inspired by the Implicit Function Theorem where G(y,p) = 0 at the optimal y.

+ To effectively perform the search, the paper introduces a two-level stereo matching network. The coarse level estimates the disparity hypothesis in coarser resolution so that the search space is reduced in refined level.

+ Experimental results show that the proposed L1 risk minimization method can improve the performance in both in-domain and cross-domain performance scenario. In addition, the ablation study also shows that the proposed L1 risk minimization can be applied for various different networks. Finally, it also show that L1-risk minimization method works better than expectation value computation to predict final disparity.

**Weaknesses:**

- The authors claim that it is better than other methods in cross-domain generalization. However, they just compare with non-domain generalization stereo matching methods. Thus, the claim is lack of the justification. It is recommended to put more analysis on the cross-domain evaluation in terms of the proposed risk minimization as done by [A] in Table 1.

- In the ablation study, the authors compare the performance of L1 risk minimization in two worst methods in Middlebury evaluation. It makes the claim to improve the performance on different networks is not justified enough. The comparison with DLNR and IGEV are also needed.

- The L1 risk minimization a general concept that can be applied into various stereo matching methods. Note that the L1 risk minimization is the biggest contribution in the paper. However, the paper introduces a new network architecture. Thus, it is unclear whether the performance in the experimental results is because of the proposed new architecture or the L1 risk minimization. More detailed analysis and ablation study are required.


Additional references:
[A] Domain Generalized Stereo Matching via Hierarchical Visual Transformation, CVPR 2023

**Questions:**

* Does the proposed risk minimization really improve the cross-domain generalization?
* Is there any reason to choose ACVNet and PCWNet for comparison in Table 8? Is it possible to perform experiment where the L1-risk is also used in the training stage. More comparisons with other networks are needed.
* If the new architecture is not used, would L1 risk minimization method improves the current state-of-the-art performance?
* Is there any limitation of the proposed L1 risk minimization method?

---

> ### Author Response · Authors · 2023-11-22
>
> ## Q1: The authors claim that it is better than other methods in cross-domain generalization. However, they just compare with non-domain generalization stereo matching methods. Thus, the claim is lack of the justification. It is recommended to put more analysis on the cross-domain evaluation in terms of the proposed risk minimization as done by [A] in Table 1.
> A1: Following your comments, we have conducted the suggested experiment showing the benefit of our method on cross-domain generalization with a recent work ITSA (Chuah et al., CVPR 2022). Statistical results related to this are provided in Table 15 in the revised draft . Despite we discussed [A] in the related work, the official source code for [A] is not public. That said, we will be happy to have a comparison with [A] once the code is available.
>
> ## Q2: In the ablation study, the authors compare the performance of L1 risk minimization in two worst methods in Middlebury evaluation. It makes the claim to improve the performance on different networks is not justified enough.
> A2: Thank you for highlighting this. This comment of yours aligns well with the advice provided by R2. Following this, we performed experiments IGEV, an iterative optimization method. We have added Table 13 in the revised draft showing the benefit of our iterative optimization approach with a well-known existing architecture in stereo matching.
>
> ## Q3: Thus, it is unclear whether the performance in the experimental results is because of the proposed new architecture or the L1 risk minimization. More detailed analysis and ablation study are required.
> A3: In this paper, we primarily  aim to propose a method to capture the continuous disparity via stereo matching networks. Furthermore, we put forward the benefit of L1 formulation to risk minimization loss function and how it can be incorporated in a differentiable way for neural network optimization via the use of implicit function theorem. Following your comments, we have more experimental results and ablation study on different architectural backbone showing the benefit of our proposed L1 risk optimization. Kindly refer Table 8, Table 13 and Table 15 for statistical results in our revised draft.
>
> ## Q4: Does the proposed risk minimization really improve the cross-domain generalization?
> A4: Yes, the experimental results show that the proposed risk minimization improved the cross-domain generalization. We have addressed this concern following your comments in Table 15 in our revised draft.
>
> ## Q5: Is there any reason to choose ACVNet and PCWNet for comparison in Table 8? Is it possible to perform experiment where the L1-risk is also used in the training stage. More comparisons with other networks are needed.
> A5: The reason we choose ACVNet and PCWNet is because those are recent feature aggregation networks that seems to work really well for stereo matching problems. Yes, it is possible to do risk optimization at train time. Yet for simplicity we used pretrained feature networks. That said this is an encouraging direction for future extension.
>
> We have added comparison with other network architecture such as IGEV. Kindly refer to table 13  in the revised draft.
>
> ## Q6: If the new architecture is not used, would L1 risk minimization method improves the current state-of-the-art performance?
> A6: We have tested our L1 risk minimization on recent off-the-shelf network designs such as IGEV, PCWNet and added its results in table 13 and table 8, respectively in the revised draft. We observed a clear improvement.
>
> ## Q7: Is there any limitation of the proposed L1 risk minimization method?
> A7: Despite its benefit, L1 risk minimization is slightly more time consuming than the popular loss function such as L2 loss. To make such a continuous optimization time efficient is an attractive avenue for research in deep learning. Kindly refer to Table 8  in our revised draft for inference time analysis.

---

> > ### Comment · Reviewer_bKMt · 2023-11-23
> >
> > I have read the reviews and the rebuttals. The authors have addressed most of the concerns of the reviewers. Risk minimization is proven to improve the performance of stereo-matching methods.

---

### Official Review · Reviewer_MXR7 · 2023-10-30

**Soundness:** 3 good
**Presentation:** 3 good
**Contribution:** 3 good
**Rating:** 6
**Confidence:** 5

**Summary:**

The authors propose to model the disparity estimation problem as a continuous risk minimization problem. They propose the use of a robust L1 formulation to account for multi-modal disparity distributions. To enable end-to-end learning they propose a differentiable scheme using the implicit function theorem for forward prediction and backward error propagation. The authors demonstrate the effectiveness of their proposed method by conducting experiments using all popular stereo vision datasets.

**Strengths:**

The method is very interesting and the results are strong. The authors have conducted experiments using all popular datasets and have compared their proposed method against many competing methods. Their experiments include in-domain evaluation, cross-domain evaluation, and ablation studies. The method is well-motivated. The authors explain their method well and they provide a good theoretical background to support their assumptions.

**Weaknesses:**

1) There are parts of the paper that wording can be improved. Proofreading by a native speaker will further strengthen the paper. Examples:
 a) impressive deep-learning architecture -> advancements in deep-learning. Something being impressive doesn't necessarily mean that it works
 b) Typical stereo matching algorithm will sample ... and compute a discrete distribution p: Typical stereo matching algorithms will compute a cost that merely can be described as a PMF.
c) To take a value from -> To choose a value as the final prediction?
d) where the Sign() is the sign function -> where the sign is the signum function?
e) In implementation, we clip -> we clip...

2) The symbols in the math equation can easily cause confusion:
   a) The symbol p is used to denote the probability mass function and the probability density function. It is common to use P for mass and F for density. Another option would be to use subscripts if the authors want to keep using the notation p(x;p) for the probability density function.
   b) in equation (2) y is in the left and right hand of the equation. If y is the result of the argmin shouldn't the right-hand y be defined as a variable in a range of values? similarly eq(4)

3) The binary search algorithm can be moved to the appendix.
4) The related work is very dense. Moving the binary search algorithm to the appendix will free some space to expand.
5) Figure 3: The captions are not aligned with the different stages of the architecture. You can condense the schematics and allow more space for risk minimization since the other parts of the proposed architecture are described in prior art
6) Table 4: Are these results after training and testing all these methods on Middlebury? Please specify in the captions of all tables if the reported results are after self-evaluation or the official results on the dataset evaluation page. Submit also on the test set of Middlebury.
7) Since the main proposal of this paper is risk minimization, It would strengthen the paper if they apply their proposed module to more architectures, like IGEV or Raft-stereo which has shown strong cross-domain generalization.

**Questions:**

N/A

---

> ### Author Response · Authors · 2023-11-22
>
> ## Q1: There are parts of the paper that wording can be improved. Proofreading by a native speaker will further strengthen the paper. Examples: a) impressive deep-learning architecture -> advancements in deep-learning. Something being impressive doesn't necessarily mean that it works b) Typical stereo matching algorithm will sample ... and compute a discrete distribution p: Typical stereo matching algorithms will compute a cost that merely can be described as a PMF. c) To take a value from -> To choose a value as the final prediction? d) where the Sign() is the sign function -> where the sign is the signum function? e) In implementation, we clip -> we clip…
> A1: We appreciate you pointing out the paper's minor typos and writing mistakes. Thank you so much for this. We have improved the highlighted typos and minor mistakes in the revised draft.
>
> ## Q2: The symbols in the math equation can easily cause confusion: a) The symbol p is used to denote the probability mass function and the probability density function. It is common to use P for mass and F for density. Another option would be to use subscripts if the authors want to keep using the notation p(x;p) for the probability density function. b) in equation (2) y is in the left and right hand of the equation. If y is the result of the argmin shouldn't the right-hand y be defined as a variable in a range of values? similarly eq(4)
> A2: Thank you for the great advice on the notation and its possible confusion. (a) We added superscript $m$ to $\mathbf{p}$, i.e. $\mathbf{p}^m$ to the notation of probability mass function to avoid the confusion. (b) We have refined the draft accordingly.
>
> ## Q3: The binary search algorithm can be moved to the appendix.
> A3: We have moved the binary search algorithm in the appendix–refer page 16 Algorithm 1.
>
> ## Q4: The related work is very dense.
> A4: We agree that related work is dense. But this is mainly because stereo matching is a very classical problem with more than 25 years of literature.  In our paper’s related work, we have tried to keep the essence of classical as well as modern approaches. We have slightly detailed on the related work in our revised draft and added a few comparison with our approach.
>
> ## Q5: Figure 3: The captions are not aligned with the different stages of the architecture. You can condense the schematics and allow more space for risk minimization since the other parts of the proposed architecture are described in prior art
> A5: Excellent advice for improving our draft. For now, we are keeping the Figure 3 illustration the same but we are open to changing it after the decision. We have changed the Figure 3 caption following your advice.
>
> ## Q6: Table 4: Are these results after training and testing all these methods on Middlebury? Please specify in the captions of all tables if the reported results are after self-evaluation or the official results on the dataset evaluation page.
> A6: For Table 4, the methods are trained on SceneFlow and evaluated on the training set of Middlebury. For Table 2 and Table 3, the results are obtained from the KITTI official website. Following your comments, we revised the captions.
>
> ## Q7: Submit also on the test set of Middlebury.
> A7: We are in the process of submitting the results on the middlebury public benchmark leaderboard and soon it will be online.
>
> ## Q8: Since the main proposal of this paper is risk minimization, It would strengthen the paper if they apply their proposed module to more architectures, like IGEV or Raft-stereo which has shown strong cross-domain generalization.
> A8: Indeed that is good advice to strengthen the current paper. We have added Table 13 in the revised draft that shows the results of our formulation on IGEV network architecture. Furthermore,  we have shown results on two other networks namely, ACVNet and PCWNet —refer Table 8 in the revised draft.

---

> > ### Comment · Reviewer_MXR7 · 2023-11-23
> >
> > I would like to thank the authors for considering the comments of the reviewers. I read the revised version of the paper and I believe that it is an improved version of the initial draft. Congratulations on your paper acceptance.

---

### Official Review · Reviewer_N1iu · 2023-11-06

**Soundness:** 3 good
**Presentation:** 3 good
**Contribution:** 3 good
**Rating:** 6
**Confidence:** 3

**Summary:**

This paper presents a novel formulation for predicting continuous disparity in stereo matching networks by framing the problem as risk minimization. Rather than directly predicting the disparity or regressing to a continuous value, the authors propose treating the prediction as an optimal solution that minimizes the risk or expected error. Under this formulation, the commonly used disparity expectation is shown to be a special L2 case. The authors then advocate using an L1 loss which is more robust, and derive a differentiable algorithm to enable end-to-end training. Experiments on multiple datasets demonstrate state-of-the-art performance and superior cross-domain generalization ability.

**Strengths:**

The risk minimization view of disparity prediction is an interesting and original perspective. Framing it as an optimization problem with an L1 loss is intuitive and theoretically sound.
The method of computing gradients for the non-differentiable L1 optimization using implicit function theorem is clever, enabling end-to-end training.
The approach convincingly outperforms current state-of-the-art methods on in-domain and cross-domain evaluations across a range of datasets. The cross-domain results in particular highlight the robustness.
The ablation studies clearly demonstrate the benefit of L1 risk minimization over disparity expectation, even without retraining networks. This underscores the general applicability.

**Weaknesses:**

While theoretically motivated, the actual network architecture and components like cost volumes seem fairly standard. More implementation details could help highlight if other architectural modifications were needed to fully take advantage of risk minimization.
The binary search algorithm for L1 optimization, while efficient, involves heuristics like the tolerance hyperparameter. More analysis could be provided on algorithm convergence and optimality guarantees.
For real-time applications, the running time and complexity should be analyzed more thoroughly. Comparisons to optimizations like pruning could better highlight efficiency.

**Questions:**

How does the risk minimization formulation compare to other robust loss functions like smooth L1 that try to reduce outlier influence? Is the improvement mainly from switching to L1 or from the risk view?
Have the authors experimented with other error functions like Huber loss within the risk framework? This could help determine the benefits of specifically L1 vs. just a robust loss.
Can the method apply to other vision tasks with multi-modal outputs like optical flow or depth estimation? Does it mainly benefit stereo matching or generalize broadly?

---

> ### Author Response · Authors · 2023-11-22
>
> ## Q1: While theoretically motivated, the actual network architecture and components like cost volumes seem fairly standard. More implementation details could help highlight if other architectural modifications were needed to fully take advantage of risk minimization.
> A1: We agree that a more insightful architectural design modification could benefit the proposed risk loss function more. However, in this paper, we primarily  aim to propose a method to capture the continuous disparity via stereo matching networks. To this end, we put forward the benefit of L1 formulation to risk minimization loss function and how it can be incorporated in a differentiable way for neural network optimization via the use of implicit function theorem. Following your comments, we have added more implementation details in A.2.
>
> ## Q2: The binary search algorithm for L1 optimization, while efficient, involves heuristics like the tolerance hyperparameter. More analysis could be provided on algorithm convergence and optimality guarantees.  For real-time applications, the running time and complexity should be analyzed more thoroughly. Comparisons to optimizations like pruning could better highlight efficiency.
> A2: Thank you for highlighting this. The argument on the tolerance hyperparameter is indeed important and generally true yet the suitable target function introduced in the paper is convex and its optimal solution can be guaranteed -refer Eq.(4) in the revised draft. Details such as O(log N) time complexity with N symbolizing the number of disparities is added in the revised draft–refer  [1] for details. Following your suggestion, we performed an experimental study in relation to the tolerance hyperparameter.  Table 11 in the revised draft shows the performance of our method with different tolerance hyperparameters across different disparity levels. In relation to pruning, it will be insightful to have a rigorous analysis of the proposed optimization and we are still to conduct more intricate and extremely detailed statistical analysis for a decisive agreement on our formulation's efficiency to sparse networks.
>
> [1] Thomas H. Cormen, Charles E. Leiserson, Ronald L. Rivest, and Clifford Stein. 2009. Introduction to Algorithms, Third Edition (3rd.
> ed.). The MIT Press.
>
> ## Q3: How does the risk minimization formulation compare to other robust loss functions like smooth L1 that try to reduce outlier influence? Is the improvement mainly from switching to L1 or from the risk view? Have the authors experimented with other error functions like Huber loss within the risk framework? This could help determine the benefits of specifically L1 vs. just a robust loss.
> A3: Thank you for this excellent suggestion. We have added Table 12 in the revised draft—refer A.4.2. that provides an ablation study showing the benefit of the proposed risk minimization with L1  in relation to the Huber loss, i.e. a combination of L1 and L2 norm depending on the thresholding value $\beta$. The table in the revised draft clearly shows the benefit of using risk minimization loss under L1.
>
> ## Q4: Can the method apply to other vision tasks with multi-modal outputs like optical flow or depth estimation? Does it mainly benefit stereo matching or generalize broadly?
> A4: This comment is very insightful. Given that depth, optical flow and disparity are correlated concepts in stereo vision, we believe our proposed loss function could provide a fruitful direction to investigate all those related problems and may also benefit similar problems in computer vision. But in this work, we primarily concern ourselves with risk minimization idea to the stereo matching problem with application to robotics and control field and showed its possible applicability accordingly.

---

### Author Response · Authors · 2023-11-22
**Thanks to all the reviewers**

We thank all the reviewers for their invaluable time and effort in reviewing our paper.
It's good to learn that all the reviewers agreed on the same score of 6, i.e., "marginally above the acceptance threshold". We are happy to note that reviewer R1's acknowledgment of the "risk minimization view of disparity prediction is an interesting and original perspective" and The cross-domain results, in particular, highlight the robustness. Additionally, R2 finds the paper's work very interesting, with strong results. On similar lines, R3 and R4 find our paper to be a novel risk of disparity concept to measure the final disparity and introduce a fresh risk minimum perspective on the challenge of capturing continuous disparity for stereo matching. Nonetheless, reviewers have minor concerns and comments to improve the draft further. We addressed those below and added suggested experiments and ablations in the revised draft.

---

### Meta-Review · Area_Chair_umEg · 2023-12-06

**Metareview:**

This paper presents a method to use L1 risk minimization which shows to be useful for stereo matching. Although the use of L1 loss for stereo matching might be new in this specific domain, it has been widely used previously in computer vision such as sparse coding. Therefore, the technical novelty is limited. The authors show some improvement in the results, but the improvements are incremental  and in many cases such as the performance in KITTI 2012 and 2015, the proposed method is not better than state-of-the-art methods such as IGEV, CroCo-Stereo. The authors also failed to provide proper justification of L1 loss compared with previous losses. The authors may also go deeper to investigate how other possible choice of losses, say L1+L2, might affect the results. As mentioned by the reviewer bKMt, it is hard to tell if the improvement (if any) comes from the network design or the loss.  A proper way shall be comparing the performances of difference losses while keeping the rest of the network the same. Then report both in-domain and cross-domain performance.  During the rebuttal, the authors have provided some results for cross-domain tests in Table 8 and 13, but how it improves the performance in in-domain test such as KITTI datasets? The visual results also fail to highlight how the L1 loss improve the results.

The title of this paper is also exaggerated as the main improvement is cross-domain performance but "robust" covers much wider concept, for example, robustness to adversarial attack. It is also hard to claim that this method is "robust stereo matching".

Based on these, I cannot support this paper.

**Justification For Why Not Higher Score:**

The paper has major weakness in validation while the technical novelty is also limited. The results show that there are some improvements for cross-domain tests, but the in-domain evaluation results are worse than state-of-the-art methods. The title of this paper is also exaggerated as the main improvement is cross-domain performance but "robust" covers much wider concept, for example, robustness to adversarial attack. It is also hard to claim that this method is "robust stereo matching".

Based on these, I cannot support this paper.

**Justification For Why Not Lower Score:**

NA

---

### Decision · Program_Chairs · 2024-01-16

Reject